# Transitional Changes in Fatigue-Related Symptoms Due to Long COVID: A Single-Center Retrospective Observational Study in Japan

**DOI:** 10.3390/medicina58101393

**Published:** 2022-10-05

**Authors:** Yasuhiro Nakano, Yuki Otsuka, Hiroyuki Honda, Naruhiko Sunada, Kazuki Tokumasu, Yasue Sakurada, Yui Matsuda, Toru Hasegawa, Kanako Ochi, Hideharu Hagiya, Hitomi Kataoka, Keigo Ueda, Fumio Otsuka

**Affiliations:** 1Department of General Medicine, Okayama University Graduate School of Medicine, Dentistry and Pharmaceutical Sciences, Okayama 700-8558, Japan; 2Center for Education in Medicine and Health Sciences, Okayama University Graduate School of Medicine, Dentistry and Pharmaceutical Sciences, Okayama 700-8558, Japan

**Keywords:** dysgeusia, dysosmia, myalgic encephalomyelitis/chronic fatigue syndrome, Omicron variant, and post COVID-19 condition

## Abstract

*Background and Objectives*: Changes in post COVID-19 condition (PCC) characteristics caused by viral variants have yet to be clarified. We aimed to characterize the differences between clinical backgrounds and manifestations in long COVID patients who were infected with the Delta variant and those who were infected with the Omicron variants. *Materials and Methods*: This study was a single-center retrospective observational study for patients who visited our COVID-19 aftercare outpatient clinic (CAC) established in Okayama University Hospital (Japan) during the period from 15 February 2021 to 15 July 2022. We classified the onset of COVID-19 in the patients into three groups, the preceding, Delta-dominant, and Omicron-dominant periods, based on the prevalent periods of the variants in our prefecture. *Results*: In a total of 353 patients, after excluding 8 patients, 110, 130, and 113 patients were classified into the preceding, Delta-dominant, and Omicron-dominant periods, respectively. Patients infected in the Omicron-dominant period had significantly fewer hospitalizations, milder illnesses, more vaccinations and earlier visit to the CAC than did patients infected in the Delta-dominant period. Patients infected in the Omicron-dominant period had significantly lower frequencies of dysosmia (12% vs. 45%, ** *p* < 0.01), dysgeusia (14% vs. 40%, ** *p* < 0.01) and hair loss (7% vs. 28%, ** *p* < 0.01) but had higher frequencies of fatigue (65% vs. 50%, * *p* < 0.05), insomnia (26% vs. 13%, * *p* < 0.05) and cough (20% vs. 7%, ** *p* < 0.01) than did patients infected in the Delta-dominant period. *Conclusions*: The transitional changes in long COVID symptoms caused by the two variants were characterized.

## 1. Introduction

More than two and a half years after the World Health Organization (WHO) declared a pandemic on 30 January 2020, coronavirus disease 2019 (COVID-19) is still raging worldwide and the cumulative number of infected people worldwide exceeded 500 million by April 2022 (World Health Organization. WHO Coronavirus (COVID-19) Dashboard; https://covid19.who.int/table, accessed on 6 August 2022). In addition to acute symptoms, COVID-19 also causes prolonged sequelae, which are referred to as post-COVID-19 conditions (PCC), post-acute sequelae of severe acute respiratory syndrome coronavirus 2 (SARS-CoV-2) (PASC), or long COVID [1]. PCC has been reported to occur in at least one-third of COVID-19 patients at 60 days after the acute-phase onset [2,3]. Even considering pre-infection physical symptoms, it has been reported that post-infection symptoms could be attributed to COVID-19 in 12.7% of the patients [4]. PCC symptoms include fatigue, dysosmia, dysgeusia, headache, hair loss, dyspnea, and insomnia [5].

Many variants of SARS-CoV-2 have emerged and have been found to have characteristics different from those of conventional strains. The Omicron variant, which WHO designated as a variant of concern on 26 November 2021, is more infectious than the Delta variant and rapidly spread throughout the world, with several sublineages also emerging [6]. A prospective observational study showed that COVID-19 patients in the acute phase who were infected during the period of Omicron variant dominance had less frequent loss of smell, more frequent sore throat, and a lower rate of hospitalization than did patients who were infected during the period of Delta variant dominance [7]. These studies indicate that the Omicron and Delta variants have different features not only in the acute phase but also in the post-infection symptoms. Furthermore, a case-control observational study showed that the rate of long COVID at 28 days after diagnosis of COVID-19 in COVID-19 patients who were infected during the Omicron-dominant period was less than half of the rate in COVID-19 patients who were infected during the Delta-dominant period (4.5% vs. 10.8%) [8]. An increased risk of developing some neuropsychiatric symptoms has been reported for the Delta and Omicron variants compared with the Alpha variant [9]. Due to the rapid spread of the Omicron variants, the number of PCC patients has been increasing rapidly, resulting in an excessive social burden.

We have recently reported clinical characteristics of PCC patients who visited our COVID-19 aftercare outpatient clinic (CAC). We found that the major sequelae were fatigue and anxiety [5,10] in the early period of the COVID-19 pandemic, that the severity of the acute infection greatly affects the persistent symptoms of PCC [11], and that some endocrine dysfunction may be involved in the fatigue symptoms of PCC [12,13]. However, the characteristics of PCC patients who were infected with the Omicron variants and the transitional changes in the variant-related symptoms have yet to be elucidated. The aim of the present study was to clarify the differences between clinical manifestations in long COVID patients who were infected during the Delta-dominant period and those in long COVID patients who were infected during the Omicron-dominant period.

## 2. Patients and Methods

### 2.1. Study Design and Patients’ Characteristics

This study was a retrospective observational study conducted in a single facility. We reviewed the medical records of patients who visited our CAC during the period from 15 February 2021 to 15 July 2022. Our CAC was established on 15 February 2021 in the Department of General Medicine, Okayama University Hospital (Japan) in order to evaluate and manage patients suffering from long COVID symptoms. Long COVID was defined as symptoms that persist for more than four weeks after the onset of COVID-19 [1,2,14]. We obtained information on age, sex, body mass index (BMI), current smoking and alcohol drinking habits, hospitalization due to COVID-19, therapeutic use of oxygen or corticosteroids during the acute phase, severity of COVID-19, history of COVID-19 vaccination (BNT162b2, mRNA-1273, or ChAdOx1), number of days between onset of COVID-19 and first CAC visit, and clinical symptoms of long COVID. The severity of the acute phase of COVID-19 was classified according to the criteria defined by the Ministry of Health, Labour and Welfare in Japan [15]. Clinical symptoms of PCC were identified through a face-to-face careful medical interview with a physician.

### 2.2. Definition of the Delta- and Omicron-Dominant Periods

We classified the onset of COVID-19 in the patients into three groups based on the epidemiological aspect of COVID-19 in Okayama Prefecture in Japan: the preceding period, Delta-dominant period, and Omicron-dominant period (Figure 1). Epidemiological information was obtained from data for the numbers of cases of new coronavirus infection, deaths, and severe cases in Japan at the website of NHK (Japan Broadcasting Corporation; https://www3.nhk.or.jp/news/special/coronavirus/data-all/, accessed on 6 August 2022) and the website of current status of mutant genome analysis in Okayama Prefecture (https://www.pref.okayama.jp/page/724270.html#04-henikabugenomukaisekijyoukyou, accessed on 6 August 2022). The preceding period is the period from the ancestral strain to the Alpha strain, before 18 July 2021; the Delta-dominant period is the period from 9 July 2021 to 31 December 2021, when the Delta variants were dominant after the Alpha-dominant phase; and the Omicron-dominant period is the period after 1 January 2022, when the Omicron variants were dominant in Okayama Prefecture, Japan.

### 2.3. Statistical Analysis

EZR, version 1.40 (Saitama Medical Center, Jichi Medical University, Saitama, Japan), which is a graphical user interface for R (The R Foundation for Statistical Computing, Vienna, Austria), was used in all statistical analyses [16]. It is modified from R commander, which is designed to add frequently used functions in biostatistics. The data were presented as number (%) for categorical variables and median (interquartile ranges: IQR) for continuous variables. We used the Mann–Whitney U test to explore differences in the variables given their non-normal distribution. We used Pearson’s χ2 test for associations between categorical variables. The threshold for significance was defined as * *p* < 0.05 and ** *p* < 0.01.

## 3. Results

Data for all of the 361 patients who visited our CAC during the study period were obtained from medical records. We excluded eight patients including two patients who visited our CAC within four weeks after onset of COVID-19, two patients who were under ten years of age, two patients who had insufficiently available data, and two patients who were asymptomatic. Clinical backgrounds of the remaining 353 patients are shown in Table 1. The eligible cases were classified into the preceding period (110 patients), the Delta-dominant period (110 patients), and the Omicron-dominant period (113 patients). There were no significant differences in age, gender, and BMI among the three groups. The percentage of subjects with a smoking habit was significantly lower in the Omicron-dominant period group than in the Delta-dominant period group (27.4% vs. 45.4%, ** *p* < 0.01).

Regarding clinical information related to COVID-19 shown in Table 1, the percentage of patients with hospital admission during the acute phase was significantly lower among patients in the Omicron-dominant period group than among patients in the Delta-dominant period group (9.7% vs. 31.5%, ** *p* < 0.01). Similarly, the proportion of patients who received oxygen and/or steroid therapy was significantly lower among patients in the Omicron-dominant period group (9.7% vs. 23.8%, ** *p* < 0.01). Additionally, large proportions of patients infected in the Omicron-dominant period had mild COVID-19 (74.6% vs. 95.6%) and had received three doses of COVID-19 vaccination (0.8% vs. 13.3%) compared to those infected in the Delta-dominant period. The duration from the onset of COVID-19 to the CAC visit was significantly shorter in the Omicron-dominant period group (median duration: 59 days vs. 95 days, ** *p* < 0.01).

The percentages of patients with each long COVID symptom at the first CAC visit in the three groups are shown in Figure 2. The most frequent symptom in all groups was fatigue (preceding period group: 54%, Delta-dominant period group: 50%, and Omicron-dominant period group: 65%). The proportion of patients with fatigue was significantly higher in the Omicron-dominant period group than in the Delta-dominant period group (* *p* < 0.05). On the other hand, dysosmia and dysgeusia, which were the second-most frequent symptoms in patients infected during the preceding period (25% and 23%, respectively) and the Delta-dominant period (45% and 40%, respectively), were significantly less frequent in patients infected in the Omicron-dominant period (12% and 14%, respectively) than in those infected in the Delta-dominant period (** *p* < 0.01 and ** *p* < 0.01, respectively). Headache and dyspnea tended to be more common in the Omicron-dominant period group (29% and 7%, respectively), but the difference was not statistically significant. In addition, the complaint of hair loss was significantly less frequent (7% vs. 28%, ** *p* < 0.01) and insomnia (26% vs. 13%, * *p* < 0.05) and cough (20% vs. 7%, ** *p* < 0.01) were significantly more frequent in patients infected in the Omicron-dominant period than in those infected in the Delta-dominant period.

## 4. Discussion

As far as we know, this study is the first study in Japan to reveal the differences between long COVID symptoms caused by the Omicron and Delta variants. We found that among the patients who visited our CAC, those who were infected during the Omicron-dominant period had lower rates of dysosmia, dysgeusia, and hair loss, but had higher rates of fatigue, insomnia, and cough than did patients who were infected during the Delta-dominant period. These results indicate that long COVID symptoms are likely to be different depending on the virus variant.

As for the fatigue symptom in PCC, we have shown that the overall prevalence rate of myalgic encephalomyelitis/chronic fatigue syndrome (ME/CFS) was 16.8% in a retrospective study on 279 patients who met the definition of PCC [10]. ME/CFS is a debilitating disorder that is characterized by fatigue, sleep disturbance, various types of pain and neurological/cognitive dysfunction persisting for more than six months [17,18,19]. Of interest, fatigue-related symptoms including dizziness, chest pain, insomnia and headache were found to be characteristic manifestations related to the progress for ME/CFS in our earlier study [10]. Therefore, the increase in PCC patients suffering from fatigue with insomnia may indicate that the Omicron-related PCC is also involved in the development of ME/CFS in the long clinical course of PCC.

Regarding respiratory symptoms, Omicron variants tend to prefer the upper airway compared to previous mutants. An ex vivo study showed that the Omicron variants proliferate more efficiently in bronchi and have lower replication competence in the lungs compared with previous lineages, and those differences could explain the higher infectivity and lower severity of the Omicron strain [20]. Patients with COVID-19 caused by the Omicron variants have been reported to have more complaints of sore throat in the acute phase [7]. Patients infected during the Omicron-dominant period in this study had a lower frequency of sore throat as a PCC symptom but had a higher frequency of cough, possibly due to prolonged pharyngeal irritation and airway hyperresponsiveness. Prolonged inflammation of the epipharynx, a lymphoid tissue in the upper airway and one of the proliferation sites of SARS-CoV-2, has been considered as one of the pathogeneses of ME/CFS [21]. It is possible that infection with the Omicron variants leads to fatigue and insomnia, which are key symptoms of ME/CFS, via some inflammatory interaction with persistent epipharyngitis [22].

Dysosmia and dysgeusia are also characteristic symptoms of the acute phase of COVID-19. Dysosmia has been reported to be less common in patients with COVID-19 caused by the Omicron variants during the acute phase [7]. The Omicron variants have more than 30 mutations in the spike protein and 15 of the mutations are in the receptor-binding domain, which could affect their infectivity and affinity for angiotensin-converting enzyme (ACE) 2 receptors [23]. Recent studies have shown that ACE2 receptors are not expressed on olfactory sensory neurons, but are present on surrounding sustentacular cells and basal cells [24]. Several reasons for the lower incidence of dysosmia due to the Omicron variants have been proposed. Possible reasons include a mechanism of entry that does not depend on transmembrane serine protease 2 (TMPRSS2), which usually plays an important role in cell invasion together with ACE2 receptors, and less inflammation induced by Omicron variants than by other variants in the olfactory system [24]. The mechanism by which SARS-CoV-2 causes dysgeusia may also be associated with dysosmia and may be related to reduced salivary flow by binding to ACE2 receptors that are highly expressed in the salivary glands [25,26].

Hair loss is also a PCC symptom associated with autoimmune and acute diseases including various viral infections. Hair loss after COVID-19 appears after approximately two months, is often telogen effluvium, and resolves after approximately five months [27,28]. A proposed hypothesis is that interleukin (IL)-6, an inflammatory cytokine, is involved in hair loss by inhibiting elongation of hair shafts and promoting regression of hair follicles [29]. The present study indicated that hair loss was less common in long COVID patients infected during the Omicron-dominant period. The manifestation of hair loss in PCC may be related to the severity of the acute phase of COVID-19 that depends on the viral variants [7].

As the COVID-19 pandemic continues, PCC has also become a major burden [30]. In the present study, it was found that patients infected in the Omicron-dominant period had less severity of acute illness such as illness requiring hospitalization or oxygen and/or steroid therapy, more vaccinations, and earlier visits to the CAC than did patients infected in the Delta-dominant period. Although management of PCC has not yet been established, early recognition of prolonged symptoms after COVID-19 as PCC and early intervention are required. This may be the reason for the shorter interval from COVID-19 onset to the CAC visit in the Omicron-dominant period group in this study.

The present study has several limitations. First, this study was a single-center retrospective study conducted in Japan. Since CAC outpatients were patients who were referred, more severe and prolonged symptoms, rather than primary or self-limiting symptoms, may have been included in this study. A multicenter prospective study is needed to examine the entire population of PCC patients. Second, we classified infections with the Omicron and Delta variants based on the prevalence period, not by genetic testing of the strains individually. The detailed characteristics based on the subtypes of Omicron variants also need to be determined in a future study. Third, the duration between the onset of COVID-19 and the CAC visit differed among the groups. This may have led us to include symptoms that might have improved earlier, especially in the Omicron-dominant period. Fourth, comorbidities in patient backgrounds were not considered; however, patients who were able to visit our CAC were included, and patients with extremely serious illnesses were therefore not included. Fifth, the precise intervals after vaccinations were not considered when the number of vaccinations was examined. Despite these limitations, our study is the first study to reveal the differences in symptoms of PCC caused by the Omicron and Delta variants.

## 5. Conclusions

Collectively, our results showed that long COVID patients infected during the Omicron-dominant period who visited our CAC had lower rates of dysosmia, dysgeusia and hair loss but had higher rates of fatigue, insomnia, and cough than did patients infected during the Delta-dominant period. These results suggest that not only the severities of acute-phase symptoms of COVID-19 but also the sequelae of long COVID are greatly affected by the vial variants. It is important to note that the manifestation of PCC could change depending on variants that emerge in the future. Further research and data accumulation are necessary to approach the pathophysiology of PCC.

## Figures and Tables

**Figure 1 medicina-58-01393-f001:**
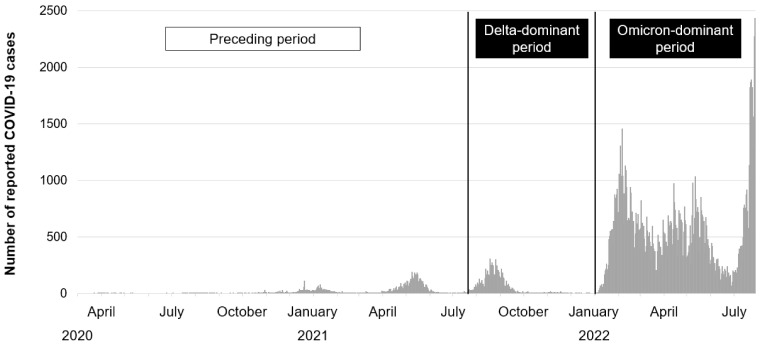
Number of reported COVID-19 cases by day in Okayama Prefecture in Japan and classification of the Delta-dominant and Omicron-dominant periods. The onset of SARS-CoV-2 viral infection in each patient with post COVID-19 condition was classified into three groups based on the epidemiological aspect of COVID-19 in Okayama Prefecture in Japan: preceding, Delta-dominant, and Omicron-dominant periods. The preceding period was the period from the ancestral strain to the Alpha strain, before 18 July 2021; the Delta-dominant period was the period from 19 July 2021 to 31 December 2021, when the Delta variants were dominant after the Alpha-dominant phase; and the Omicron-dominant period was the period after 1 January 2022, when the Omicron variants were dominant in Okayama Prefecture, Japan.

**Figure 2 medicina-58-01393-f002:**
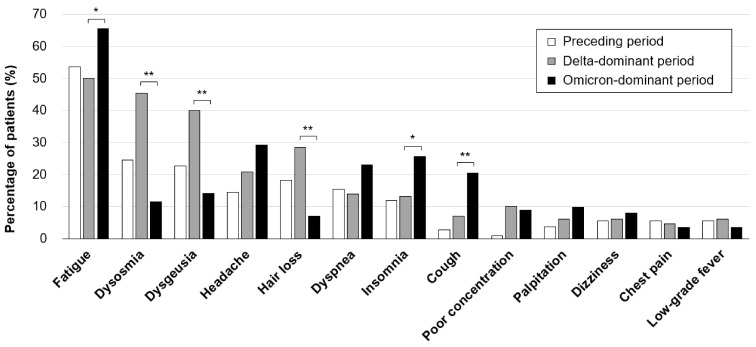
Proportions of long COVID symptoms in patients infected in the preceding, Delta-dominant, and Omicron-dominant periods. The percentages of major PCC symptoms in the patients infected during the preceding, Delta-dominant, and Omicron-dominant periods (preceding period: *n* = 110; Delta-dominant period: *n* = 130; Omicron-dominant period: *n* = 113) are shown. The χ2 test was performed for the Delta- and Omicron-dominant period groups. We regarded * *p* < 0.05 and ** *p* < 0.01 as statistically significant differences between the Delta- and Omicron-dominant period groups.

**Table 1 medicina-58-01393-t001:** Clinical characteristics of 353 patients with long COVID.

Infected Periods	Preceding Period	Delta-Dominant Period	Omicron-Dominant Period	*p* Value (Delta vs Omicron)
Number of patients	110	130	113	
Age (years), median (IQR)	43 (29–56.8)	40 (24.3–48)	39 (26–49)	0.86
Sex				
Male	48 (43.6%)	55 (42.3%)	57 (50.4%)	0.25
Female	62 (56.4%)	75 (57.7%)	56 (49.6%)	
BMI, median (IQR)	23.7 (20.5–27.0)	23.0 (20.5–25.7)	22.2 (20.3–26.0)	0.62
Smoking habit	40 (36.4%)	59 (45.4%)	31 (27.4%)	** <0.01
Alcohol drinking habit	48 (43.6%)	61 (46.9%)	55 (48.7%)	0.73
Clinical condition in acute phase				
Hospital admission	40 (36.4%)	41 (31.5%)	11 (9.7%)	** <0.01
O_2_ and/or steroid therapy	32 (29.1%)	31 (23.8%)	11 (9.7%)	** <0.01
Severity				
Mild	74 (67.2%)	97 (74.6%)	108 (95.6%)	
Moderate	32 (29.1%)	31 (23.8%)	5 (4.4%)	
Severe	4 (3.6%)	2 (1.5%)	0 (0%)	
COVID-19 vaccination				
None	74 (67.3%)	58 (44.6%)	26 (23.0%)	
1 dose	3 (2.7%)	23 (17.7%)	1 (0.9%)	
2 doses	25 (23.7%)	45 (34.6%)	70 (61.9%)	
3 doses	7 (6.4%)	1 (0.8%)	15 (13.3%)	
Days after onset to visit, median (IQR)	106.5 (70.3–211.8)	95 (63–128.8)	59 (45–87)	** <0.01

Medians [IQR: interquartile ranges] and percentages (%) are shown. BMI: body mass index, COVID-19: coronavirus disease 2019. The Mann-Whitney U test and χ2 test were performed for the Delta- and Omicron-dominant period groups. We regarded ** *p* < 0.01 as statistically significant differences between them.

## Data Availability

Detailed data will be available if requested from the corresponding author.

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
