# Peer review of "Transitional Changes in Fatigue-Related Symptoms Due to Long COVID: A Single-Center Retrospective Observational Study in Japan"

_medicina, 2022, doi:10.3390/medicina58101393_

Round 1

Reviewer 1 Report

Dear authors, thank you for the conducted study and well-written manuscript.

I have only several minor concerns.

1.       In accordance with the general requirements, it would be better to mention in the title the type of study and its location (Japan).

2.       Please add information about the types of vaccine used, whether they were from the same manufacturer.

3.       Unfortunately, I could not find whether there were people among the patients who fell ill with COVID again.

4.       Finally, given the specifics of the journal, it would be fine to see at the end of the article your recommendations (ideas) for healthcare professionals that would be interesting and useful for both clinical and research workers.

Author Response

Response to the Reviewers' Comments:

Reviewers' comments:

Reviewer #1: 
Dear authors, thank you for the conducted study and well-written manuscript.
I have only several minor concerns.

…Answer: 
We are very grateful for the reviewer’s evaluation of our manuscript.  The comments provided by the reviewer were very constructive and important to discuss our data.  We revised our manuscript according to the reviewer’s comments and suggestions.  Please refer to our revised version.

1. In accordance with the general requirements, it would be better to mention in the title the type of study and its location (Japan).

…Answer: 
Thank you for your constructive comments.  We have added “a single-center retrospective observational study in Japan” in the title.

2. Please add information about the types of vaccine used, whether they were from the same manufacturer.

…Answer: 
Thank you for your important comment.  Although we collected information on the number of vaccine doses, we did not confirm the exact types of vaccine used.  At that time, there were only three types of vaccines available in Japan, so we listed the types of vaccine (BNT162b2, mRNA-1273, or ChAdOx1) in Patients and Methods.

3. Unfortunately, I could not find whether there were people among the patients who fell ill with COVID again.

…Answer: 
Thank you for your important comment.  Unfortunately, we have not confirmed whether the patients were reinfected or not.  Although we cannot exclude the possibility that the manifestation of sequelae may differ depending on reinfection, we considered initial infection and reinfection to be the same pathology in this study.

4. Finally, given the specifics of the journal, it would be fine to see at the end of the article your recommendations (ideas) for healthcare professionals that would be interesting and useful for both clinical and research workers.

…Answer: 
Thank you for your constructive comments.  This study revealed that the symptoms of PCC can vary depending on the variants.  Therefore, we should recognize that the manifestation of PCC could change depending on variants that emerge in the future.  We have added this in the revised text.

Thank you for your review.

Reviewer 2 Report

Dear authors,

I have read with great interest your manuscript. It is well-written and easy to follow. However, I have some recommendations:

-the Introduction should be expanded

-for an original article it should be at least 30 references. Please add more.

-add an abbreviation subtitle at the end of the article

Author Response

Response to the Reviewers' Comments:

Reviewers' comments:

Reviewer #2: 
Dear authors, I have read with great interest your manuscript. It is well-written and easy to follow. However, I have some recommendations:

…Answer: 
We are very grateful for the reviewer’s evaluation of and interest in our manuscript.  The comments provided by the reviewer were very constructive and important to discuss our data.  We revised our manuscript according to the reviewer’s comments and suggestions.  Please refer to our revised version.

-    the Introduction should be expanded

…Answer: 
Thank you for your important comment.  We have expanded the Introduction by mentioning recent reports on the epidemiology of PCC and the differences in some PCC symptoms by variants.

-    for an original article it should be at least 30 references. Please add more.

…Answer: 
Thank you for your important comment.  We have added three references, for a total of 30 references cited.

-    add an abbreviation subtitle at the end of the article.

…Answer: 
Thank you for your constructive comment.  We have added subtitle “Post COVID-19 condition due to Omicron variants” at the end of the 1st page of article.

Thank you for your review.

Reviewer 3 Report

The research paper addresses a very interesting research topic concerning the characteristics of PCC in different SARS-CoV2 variants. The strength of the study are a notable number of patients seen at a single institution with very recent data presented in the retrospective study (until July 2022).

Major concerns are:

(1) In the study population Post Covid Condition (PCC) is defined as "symptoms that persist for more than four weeks after the onset of Covid-19" (page 2, line 70-71). One of the references cited is Soriano et al., 2022 which describes the Delphi consensus of  a case definition for PCC by WHO. The case definition of PCC according to WHO is: persisting symptoms after three months lasting for at least two months. This definition should be applied consistently in studies/ papers to  facilitate comparibility.   

(2) As stated in the paper the duration "from the onset of Covid-19 to the CAC visit was significantly shorter in the Omicron-dominant period group (median 59 vs 95 days)" (page 4, line 140-41). Therefore the comparison of the symptoms is problematic as symptoms and symptom severity can change dramatically during the first weeks/ months after Covid-19 infection. Adding the fact that the PCC case definition according to WHO is persistent symptoms three months after Covid-19 infection (see 1) this is a major weekness of the study as the median duration in the Omicron-dominant subgroup is only 59 days. The comparison should be done at the same duration after Covid-19 infection. 

A minor remark:

(1) Page 6, lines 179-188 refers to patients with ME/ CFS after Covid-19 infection. It should be made clear that the diagnosis of ME/ CFS can only be established when symptoms persist for six months.  

Author Response

Response to the Reviewers' Comments:

Reviewers' comments:

Reviewer #3: 
The research paper addresses a very interesting research topic concerning the characteristics of PCC in different SARS-CoV2 variants. The strength of the study are a notable number of patients seen at a single institution with very recent data presented in the retrospective study (until July 2022).

…Answer: 
We are very grateful for the reviewer’s evaluation of and interest in our manuscript.  The comments provided by the reviewer were very constructive and important to discuss our data.  We revised our manuscript according to the reviewer’s comments and suggestions.  Please refer to our revised version.

Major concerns are:
(1) In the study population Post Covid Condition (PCC) is defined as "symptoms that persist for more than four weeks after the onset of Covid-19" (page 2, line 70-71). One of the references cited is Soriano et al., 2022 which describes the Delphi consensus of a case definition for PCC by WHO. The case definition of PCC according to WHO is: persisting symptoms after three months lasting for at least two months. This definition should be applied consistently in studies/ papers to facilitate compatibility.

…Answer: 
Thank you for your important comments.  Indeed, we know that the WHO definition of PCC is symptoms lasting more than two months out of three months after COVID-19.  However, many patients visit our CAC after one month or more, and subacute symptoms are also clinically problematic when considering real data.  In addition, to date we have reported many papers based on symptoms at four weeks after COVID-19.  Since we have collected data based on this definition in the present study, we used “long COVID” instead of “PCC” in the revised manuscript.

(2) As stated in the paper the duration "from the onset of Covid-19 to the CAC visit was significantly shorter in the Omicron-dominant period group (median 59 vs 95 days)" (page 4, line 140-41). Therefore, the comparison of the symptoms is problematic as symptoms and symptom severity can change dramatically during the first weeks/ months after Covid-19 infection. Adding the fact that the PCC case definition according to WHO is persistent symptoms three months after Covid-19 infection (see 1) this is a major weakness of the study as the median duration in the Omicron-dominant subgroup is only 59 days. The comparison should be done at the same duration after Covid-19 infection.

…Answer: 
Thank you for your important comments.  We know that it is most ideal to compare symptoms in the same duration after COVID-19, but it is also true that the interval between onset and visit to our CAC is becoming shorter as PCC becomes more publicly known, and we believe that it is clinically meaningful to compare symptoms at the time of the first visit.  However, because it must be a major limitation, we have added this as a limitation in the revised discussion.

A minor remark:
(1) Page 6, lines 179-188 refers to patients with ME/ CFS after Covid-19 infection. It should be made clear that the diagnosis of ME/ CFS can only be established when symptoms persist for six months.

…Answer: 
Thank you for your important comments.  We have added in the text that ME/CFS is a disease that is characterized by symptoms such as fatigue persisting for more than six months.

Thank you for your review.

Round 2

Reviewer 3 Report

The revised version of the manuscript is well written and was improved significantly. The aouthors addressed all comments and suggestions of the first review. Follow up data on this study population would be of great interest to gain better understanding of PCC in different subgroups.